# On the Error Resistance of Hinge Loss Minimization

**Kunal Talwar**
Apple*
Cupertino, CA 95014
`ktalwar@apple.com`

## Abstract

Commonly used classification algorithms in machine learning, such as support vector machines, minimize a convex surrogate loss on training examples. In practice, these algorithms are surprisingly robust to errors in the training data. In this work, we identify a set of conditions on the data under which such surrogate loss minimization algorithms provably learn the correct classifier. This allows us to establish, in a unified framework, the robustness of these algorithms under various models on data as well as error. In particular, we show that if the data is linearly classifiable with a slightly non-trivial margin (i.e. a margin at least $C/\sqrt{d}$ for $d$-dimensional unit vectors), and the class-conditional distributions are near isotropic and logconcave, then surrogate loss minimization has negligible error on the uncorrupted data even when a constant fraction of examples are adversarially mislabeled.

## 1 Introduction

A commonly used paradigm in supervised learning is to minimize a surrogate loss over available training examples. In other words, to learn the parameters $\mathbf{w}$ of a classification model, we optimize a loss function of the form $\sum_i \ell(\mathbf{w}, \mathbf{z}_i)$ over available labeled training examples $\{\mathbf{z}_i\}$. Often the parameters $\mathbf{w}$ are themselves constrained to be in a certain set, or regularized. This paradigm has been extremely successful and underlies most applications of supervised learning.

The training examples can come from varying sources. Often, several of the training examples are mislabeled. This could be due to some inherent noise in the process, or due to adversarial mislabeling. For example, when learning a spam filter, one may use training examples labeled by users and some of these users may be spammers that insert training examples to make the system behave a certain way. These issues of noisy data, or data poisoning attacks are not new to machine learning. They have been explored in statistics under the name robust statistics [Huber, 1964, Huber and Ronchetti, 2009, Hampel et al., 2011], and in learning theory under various models of corruption [Valiant, 1985, Kearns and Li, 1988, Angluin and Laird, 1988]. In this work, we will largely be interested in the adversarial label corruption model, where the adversary can flip the labels of an arbitrary $\eta$ fraction of the examples.

There are at least two possible ways in which label corruptions might occur. The first is model misspecification: the true data distribution may not be linearly classifiable given the features. In this case, we should aim to minimize the error rate on the whole distribution. A different reason, and the focus of this work, is where the primary source of corruption is noisy, or adversarially corrupted labels. We find this to be a natural model in many training settings, where some of the data comes from users, e.g. labels coming from CAPTCHAs or from "Report Spam"/"Report Inappropriate Content" buttons, where some of these labels will come from bots or spammers. Recent works have studied this model in the stochastic bandit setting [Lykouris et al., 2018, Gupta et al., 2019]. In

such cases, we have errors in the provided training data, but the goal is to do well on uncorrupted distribution coming from real users. and ignore the performance on inputs from bots/spammers.

The problem of robustly learning a linear classifier (under adverarial label corruption) is surprisingly hard in the worst case. For proper learning, i.e. where we want the learnt classifier to be linear as well, it is hard to approximate the error rate [Arora et al., 1997, Guruswami and Raghavendra, 2009, Feldman et al., 2006] to a multiplicative factor better than $2^{\log^{1-\epsilon} d}$. Under slightly stronger complexity assumptions, a similar hardness holds for arbitrary learning algorithms [Daniely et al., 2014, Daniely, 2016]. Thus we have little hope of designing robust algorithms for learning linear classifiers in the worst case.

There is a large body of work on designing efficient, robust algorithms that work under additional assumptions on the data and on the outliers[2]. One line of work relaxes the distribution-independent PAC model, and studies specific nicer distributions, such as the uniform distribution over the unit ball. Under such assumptions, one can get arbitrarily close to the underlying corruption rate in polynomial time.

The assumptions, while natural, are arguably too restrictive. For example, data distributions of interest often have a margin, whereas isotropic distributions studied in previous work are incompatible with a reasonable margin. This motivates the question: *What conditions on the data distribution allow for efficient robust learning of linear classifiers?*

Another line of work looks at more constrained models of corruption. E.g. a recent work of Diakonikolas et al. [2019] shows that under the Massart Noise model, one can efficiently learn in the PAC model with error rate arbitrarily close to the noise rate.

These results, using sophisticated algorithms, serve to explain why it should be possible to efficiently learn outlier-robust linear classifiers. In practice, algorithms such as SVMs and logistic loss minimization are usually used and seem to be surprisingly robust to outliers. This begs the question: *Can we explain the robustness of these methods?*

Additionally all existing works in the malicious noise model aim to learn a classifier that has error rate (on inliers and outliers) close to the corruption rate $\eta$. When we care about the error rate on the inliers alone, this leads to an error rate close to $\eta/(1-\eta)$. This is information-theoretically optimal without additional assumptions. In this work, we ask: *Under what assumptions can we bypass this lower bound and get error-rates smaller than the corruption rate, on the inlier distribution?*

In this work, we address these three questions. We identify a set of conditions under which minimizing a surrogate loss allows us to learn a good classifier even in the presence of outliers. We assume that the inlier distribution is separable with a margin. Our general results (see Section 1.1) allow us to derive corollaries for different data and noise models. For example, we prove the following result.

**Theorem 1.** *(Informal) Suppose that the data distribution is supported on the unit ball in $\mathbb{R}^d$, and there is a linear separator with margin $\gamma = \Omega(\frac{\log d}{\sqrt{d}})$. Suppose that the positive and negative example distributions are mixtures of $O(1)$ isotropic log-concave distributions, each with means having norm $O(\gamma)$. There is a constant $\eta$ such that for adversarially corrupted label error rate up to $\eta$, the hinge loss minimizer on a $\tilde{Theta}(d)$-sized sample has error rate $\frac{1}{poly(d)}$ on the original data distribution.*

The data distribution assumption here is perhaps the simplest data distribution that is compatible with the margin condition. We show that a constant fraction of adversarial label errors can be tolerated while getting accuracy close to 1.

Our approach is motivated by works on "beyond worst-case analysis" [Candes and Tao, 2005, Ostrovsky et al., 2012, Bilu and Linial, 2012, Balcan et al., 2013, Awasthi et al., 2010b,a, Voevodski et al., 2010]. We identify a set of deterministic conditions under which minimizing a surrogate loss allows us to learn a good classifier even in the presence of outliers. We then show that under various models for data and noise, the conditions hold with appropriate parameters, which allows us to establish robustness. Our assumptions are weaker than the distributional assumptions in previous work. We make an additional assumption of the inliers being separable with a margin.

## 1.1 Results and Techniques

We will work with examples $(\mathbf{x}_i, y_i)$ where $\mathbf{x}_i \in \mathbb{R}^d$ and has norm at most 1, and $y_i \in \{+1, -1\}$. We assume that the *inliers* are correctly classified by a linear classifier $\mathbf{w}^\star$. In fact, we will assume **margin-separability**, which says that there is $\mathbf{w}^\star$ with norm at most $\frac{1}{\gamma^\star}$ that satisfies $y_i \mathbf{x}_i^\top \mathbf{w}^\star \geq 1$ for all inliers. Geometrically, this says that there are no points in a band of width $\approx \gamma^\star$ around the hyperplane defined by $\mathbf{w}^\star$. Such margin assumptions are standard in learning literature.

A new condition that we introduce is the **dense pancakes condition**. Informally, this says that if we project all inliers onto any direction $\mathbf{w}$, then most points are not too isolated from other inliers. Geometrically, this says that for a point $\mathbf{x} \in \mathbb{R}^d$, a "pancake" around $\mathbf{x}$, i.e. the set $\{\mathbf{x}' \in \mathcal{X} : \mathbf{w}^\top \mathbf{x} - \tau \leq \mathbf{w}^\top \mathbf{x}' \leq \mathbf{w}^\top \mathbf{x} + \tau\}$, is sufficiently dense, i.e. contains a $\rho$ fraction of the inliers. We require that for all directions, most pancakes are $\rho$-dense (the parameter $\rho$ can be arbitrary and affects our tolerance to outliers). The precise definition is slightly more complex and deferred to Section 2.

Finally, the relevant measure of the effect of the outliers in our work is the norm of the sum of (a subset of) the examples $\mathbf{x}_i$. For a set of examples $O$, we define their **Hereditary Sum Norm** HerSumNorm($O$) as $\max_{O' \subseteq O} \| \sum_{(\mathbf{x}_i, y_i) \in O'} y_i \mathbf{x}_i \|$.

Under assumptions on these parameters, we show the following theorem.

**Theorem 2.** *(Informal) Let $\mu$ be a $\gamma^\star$-margin separable distribution and let $I$ be a sample of $(1 - \eta)n$ examples drawn from $\mu$. Let $O$ be an arbitrary dataset of $\eta n$ examples in $\mathcal{X} \times \{-1, 1\}$ and let $D = I \cup O$. Let $\mathbf{w}$ be an appropriately constrained optimum for the hinge loss on $D$. If $(D, \mu)$ satisfies the $(\tau, \rho, \beta)$-dense pancakes condition for $\rho, \beta \in (0, 1)$, with $\tau \leq \gamma^\star/2$ and $(1 - \eta)\rho\gamma^\star n > 2\text{HerSumNorm}(O)$, then $\mathbf{w}$ has accuracy at least $(1 - \beta)$ on $\mu$.*

This result defines a recipe for proving robustness results for various combinations of assumptions on inlier distribution and assumptions on the corruption. Besides, the margin, the relevant ingredients are simply the density of the pancakes in inliers, and sum norm bound for the corrupted data points.

We then develop tools to establish these conditions under different models. We first study the pancakes condition. We show that whenever the data distribution is isotropic and logconcave, we can establish the dense pancake condition. Further, the pancakes condition is robust enough to easily handle translation, mixing and homogenization transformations.

We next investigate the SumNorm condition for the outliers. We study several noise models. In the malicious noise model, where the outliers are arbitrary, the best bound one can prove on the HerSumNorm is linear in the number of outliers (and this is tight). This gives us tolerance to an $\Omega(\gamma^\star)$ fraction of malicious outliers.

In a slightly more constrained noise model, where the adversary can change the labels but not the points themselves, the situation improves dramatically. In this case, we show that for any isotropic logconcave distribution, the HerSumNorm is in fact bounded by approximately $|O|/\sqrt{d}$, even if the points whose labels are corrupted are adversarially chosen. This is because even though we are adding up to $|O|$ unit vectors, they will in general not be aligned and the projection in any fixed direction is only about $1/\sqrt{d}$. This allows us to show that if the margin is at least $C/\sqrt{d}$ for a large enough constant $C$, then a constant fraction of labels can be adversarially flipped with virtually no effect on the accuracy of the learnt classifier!

We note that in our results, the learnt classifier has accuracy at least $1 - \beta$ on the inlier distribution, where $\beta$ depends only on the pancakes condition and can be much smaller than the error rate $\eta$. This is in contrast to most previous work on agnostic learning in the distributional model (see Section 1.2). The additional margin assumption we make allows us to prove this much stronger form of robustness. We remark that the margin assumption is only needed for what we call inliers. If an $\alpha$ fraction of the true inliers violate the margin assumption, they can be considered as outliers, increasing $\eta$ by $\alpha$. Our result would then apply and give an overall error rate of $(\alpha + \beta)$ on the actual inlier distribution.

Other than the benefit that we are able to analyze commonly-used algorithms, our approach offers an additional advantage. Since we have a deterministic condition that implies the robustness, the result is robust to some changes in the data distribution. For example, we show that the pancake condition is preserved under translations, and approximatley preserved under mixing of distributions. Thus if the class-conditional distributions are each a uniform mixture of a few isotropoic logconcave

distributions, then the pancake condition continues to hold. The sumnorm condition is similarly robust.

Unlike most previous work on properties of surrogate loss minimization, our result is not based on controlling the objective function value, but rather depends on the (first-order) optimality conditions. Our conclusion about correct classification is not based on the loss being small; in fact the loss itself can be large on many examples. Our proof relates the optimality conditions to the 0-1 loss of the resulting classifier.

Our theory applies to the $\ell_2$ geometry, but one can envision version of our theorems for $\ell_p$ for other $p$'s. The $\|\cdot\|_1$-$\|\cdot\|_\infty$ case, where $\mathbf{w}$ is regularized in the $\ell_1$ norm, is a particularly compelling research direction. While our main result would technically extend to Kernel methods, our current approach to infer the dense pancakes condition on the empirical sample requires the dataset size to to be $\Theta(d)$, making it inapplicable to the Kernel setting. While one can use random projections to $\Theta(\frac{1}{\gamma^2})$ dimensions and apply the algorithm in the projected space, extending our results to the usual SVM with Kernels is an interesting open question.

The rest of the paper is organized as follows. We present next additional related work. In Section 2, we set up notation and define the dense pancakes condition as well as HerSumNorm. Section 3 proves that our conditions, for appropriate parameters, imply correct classification. We derive results for Adversarial Label Noise in Sections 4 and 5. Additional tools for proving the pancakes condition, other noise models, and all missing proofs can be found in the Supplementary material.

## 1.2 Related Work

There is a long line of work on learning halfspaces under uniform or log-concave distributions under the agnostic noise model [Kalai et al., 2008, Awasthi et al., 2014, Daniely, 2015], as well as malicious noise model [Klivans et al., 2009]. Awasthi et al. [2015, 2016] study the problem under the Massart noise model, and show that for isotropic log-concave distribution, these can be learnt to arbitrarily small error $\varepsilon$ for corruption rate $\eta < 1/2$.

Long and Servedio [2011] study the margin-separable problem in the PAC model and show an algorithm that can tolerate $\eta = \Omega(\varepsilon\gamma\sqrt{\log\frac{1}{\gamma}})$, i.e. the learnt classifier has error rate $\varepsilon$ for margin $\gamma$ as long as the corruption rate is at most $\eta$. Interestingly, they also show that any minimizer of a convex surrogate can only tolerate $\eta = O(\varepsilon\gamma)$. Servedio [2003] previously showed that an online variant of Perceptron already achieves this bound. Long and Servedio [2010] showed that random classification noise already makes a large class of convex boosting-type algorithms fail. Ben-David et al. [2012] similarly showed that any convex surrogate can be made to fail badly in the absence of a margin, even with a small error rate. Moreover, they studied the margin-separable case and showed that the hinge loss is close to optimal in the worst case and can tolerate $\eta = \Omega(\varepsilon\gamma)$. Hinge-loss minimization is also used as a subroutine in Zhang [2018] for learning a sparse classifier for isotropic log-concave distributions under certain noise models.

In the non-robust setting, Bartlett et al. [2006] compare various convex surrogates in terms of consistency. There has also been a lot of recent interest in robust learning in distributional models, where some fraction of the data can be adversarial (e.g [Diakonikolas et al., 2017, 2016, 2018, Charikar et al., 2017, Klivans et al., 2018]). Compared to our work, they have weaker assumptions, but their bounds on inlier error are much worse, and hold for more complex algorithms.

## 2 Preliminaries

In this work, we will be dealing with binary classification over $\mathbb{R}^d$ using a linear classifier. We will be restricting ourselves to examples $\mathbf{x}_i \in \mathcal{X}$, where $\mathcal{X} = \{\mathbf{x} \in \mathbb{R}^d : \|\mathbf{x}\|_2 \leq 1\}$ is the the Euclidean unit ball. In the rest of the paper, $\|\cdot\|$ denotes the $\ell_2$ norm unless otherwise stated. Each example $(\mathbf{x}_i, y_i)$ has a label in $\{+1, -1\}$. A vector $\mathbf{w} \in \mathbb{R}^d$ defines a linear classifier[3] as $\text{sgn}(\mathbf{w}^\top \mathbf{x})$. We say $\mathbf{w}$ correctly classifies $(\mathbf{x}, y)$ if $y = \text{sgn}(\mathbf{w}^\top \mathbf{x})$.

We will denote by $\mathcal{D}$ a distribution over $\mathcal{X} \times \{-1, +1\}$. An empirical sample $D_n$ from $\mathcal{D}$ will consist of $n$ independent samples from a distribution $\mathcal{D}$. We next define the notion of margin separability.

**Definition 3** (Margin-Separability). *Let D be a dataset $\{(\mathbf{x}_i, y_i)\}_{i=1}^n$ where $\mathbf{x}_i \in \mathcal{X}$ and $y_i \in \{-1, 1\}$. We say that D is $\gamma^\star$-margin separable by a vector $\mathbf{w}^\star \in \mathbb{R}^d$ if $y_i \mathbf{x}^\top \mathbf{w}^\star \geq 1$ for all i and $\|\mathbf{w}^\star\| = \frac{1}{\gamma^\star}$*

**Surrogate Loss Minimization:** A common approach to practically finding a good linear classifier for a distribution is to find one that minimizes a surrogate loss on the samples. For the case of linear classifiers, a common surrogate loss is the *hinge loss* $\ell^{\text{hinge}}$ defined as:

$$\ell^{\text{hinge}}(\mathbf{w}; \mathbf{x}_i, y) \overset{eq}{=} \max(0, 1 - y_i \mathbf{w}^\top \mathbf{x}_i).$$

More generally, we will allow a larger class of loss functions.

**Definition 4.** *Let $f : \mathbb{R} \to \mathbb{R}$ be continuous and satisfy the following conditions for some subderivative[4] $f'$:*

  *1. f is non-increasing, i.e. $f'(x) \leq 0$ for all $x \in \mathbb{R}$.*

  *2. $|f'|$ is upper bounded, i.e. $f'(x) \geq -L$ for all $x \in \mathbb{R}$.*

  *3. $f'(x) \leq -1$ for all $x \leq \frac{1}{2}$.*

  *4. $f'(x) = 0$ for all $x \geq 1$.*

*Then we call the loss function $\ell^f(\mathbf{w}; \mathbf{x}, y) \overset{eq}{=} f(y\mathbf{w}^\top \mathbf{x})$ L-admissible.*

It is immediate that the hinge loss $\ell^{\text{hinge}}$ is 1-admissible as the step function $g(x) = -\mathbb{1}(x \leq 1)$ is a subderivative satisfying all the conditions of the definition. The logistic loss satisfies the first three properties and can be truncated (i.e. $\ell(x) = \max(\ell^{\text{logistic}}(x), \ell^{\text{logistic}}(1))$) to satisfy the fourth property as well. For a parameter $\gamma$ and an admissible loss function $\ell$, we consider the following optimization problem:

$$\min_{\mathbf{w}} \quad \sum_i \ell(\mathbf{w}; \mathbf{x}_i, y_i)$$
$$\text{s.t.} \quad \|\mathbf{w}\|_2 \leq \frac{1}{\gamma}. \tag{$SLM_{\ell,\gamma}$}$$

Note that $f$ and thus the loss function may be non-convex. Our result will then hold for any first-order critical point of the empirical loss.

**Error model:** We will consider a setting where the dataset is comprised of some *inliers I* that will be margin-separable, and some *outliers O* that may be mislabeled by $\mathbf{w}^\star$. For simplicity, one can think of the outliers as being produced as a result of an adversary corrupting a certain fraction of labels; we consider various error models in Section 5. Let $n$ denote the total number of examples in the dataset. We will restrict the corruption to an $\eta$ fraction of the points, so that $|I| \geq (1 - \eta)n$ and $|O| \leq \eta n$.

**Dense Pancakes Condition:** We next define the dense pancakes condition. Informally, the condition stipulates that points are not too isolated from other points, though the precise condition is significantly weaker.

**Definition 5.** *Let $(\mathbf{z}, y) \in \mathcal{X} \times \{-1, +1\}$ and let $\mu$ be a measure on $\mathcal{X} \times \{-1, +1\}$. For a unit vector $\mathbf{w} \in \mathbb{R}^d$, the $\mathbf{w}$-pancake of width $\tau$ at $(\mathbf{z}, y)$ is defined as the set $P_{\mathbf{w}}^\tau(\mathbf{z}, y) = \{(\mathbf{x}, y_\mathbf{x}) \in \mathcal{X} \times \{-1, +1\} : y\mathbf{w}^\top \mathbf{z} - \tau \leq y_\mathbf{x} \mathbf{w}^\top \mathbf{x} \leq y\mathbf{w}^\top \mathbf{z} + \tau\}$. We say that the pancakes $P_{\mathbf{w}}^\tau(\mathbf{z}, y)$ is $\rho$-dense with respect to $\mu$ if $\mu(P_{\mathbf{w}}^\tau(\mathbf{z}, y)) \geq \rho$.*

**Definition 6.** *We say that a pair of distributions $(\mu, \nu)$ satisfies the $(\tau, \rho, \beta)$-dense pancakes condition if for every unit vector $\mathbf{w} \in \mathbb{R}^d$, the pancake $P_{\mathbf{w}}^\tau(\mathbf{z}, y)$ is $\rho$-dense w.r.t. $\mu$, except with probability $\beta$, when $(\mathbf{z}, y)$ is drawn from $\nu$. Formally:*

$$\forall \mathbf{w} \in \mathbb{R}^d, \|\mathbf{w}\| = 1 \quad : \quad \Pr_{(\mathbf{z}, y) \sim \nu}[P_{\mathbf{w}}^\tau(\mathbf{z}, y) \text{ is } \rho\text{-dense w.r.t. } \mu] \geq 1 - \beta$$

Sometimes, we will abuse notation and use a finite set $D$ to mean the uniform distribution $\mu_D$ over it. We will also use the shorthand "$\mu$ satisfies the $(\tau, \rho, \beta)$ dense pancakes condition" to mean that "$(\mu, \mu)$ satisfies the $(\tau, \rho, \beta)$ dense pancakes condition".

**Sum Norm:** The following definition would be useful in stating the theorem.

**Definition 7.** *Let $D \subseteq \mathcal{X} \times \{-1, +1\}$ be finite. The* Sum Norm *of the set, denoted by* SumNorm$(D)$, *is defined as* $\|\sum_{(\mathbf{x}_i, y_i) \in D} y_i \mathbf{x}_i\|$. *The* Linear Sum Norm, *denoted by* LinSumNorm$(D)$, *is defined by the expression* $\sup_{a_1, \ldots, a_{|D|} : 0 \leq a_i \leq 1} \|\sum_{(\mathbf{x}_i, y_i) \in D} a_i \mathbf{x}_i\|$.

Note the by triangle inequality, SumNorm$(D) \leq$ LinSumNorm$(D) \leq |D|$. A related notion will often be easier to work with:

**Definition 8.** *Let $D \subseteq \mathcal{X} \times \{-1, +1\}$. The* Hereditary Sum Norm *of the set, denoted by* HerSumNorm$(D)$, *is defined as* $\sup_{D' \subseteq D}$ SumNorm$(D')$.

The following elementary lemma shows why it suffices to control the HerSumNorm. We defer the proof to Appendix E.

**Lemma 9.** *Let $D \subseteq \mathcal{X} \times \{-1, +1\}$ Then* HerSumNorm$(D) =$ LinSumNorm$(D)$.

In our work, LinSumNorm abstracts exactly the property of the outliers that is needed for the proof, as it bounds their contribution to the gradient.

# 3 The Dense Pancakes Lemma

We will show that any point $\mathbf{z}$ that satisfies a suitable dense pancakes condition will not be misclassified by a solution to $(SLM_{\ell, \gamma})$ for suitable parameters. The basic intuition of the proof is as follows: the first order optimality conditions imply that a weighted sum of example gradients at the optimum is close to zero. A dense pancake around a misclassified point gives us a sufficiently large sum of gradients which must be balanced by the contribution from the outliers.

**Lemma 10** (Dense Pancakes Lemma)**.** *Let $I$ be a dataset containing $(1 - \eta)n$ examples and suppose that $I$ is $\gamma^\star$-margin separable by a classifier $\mathbf{w}^\star$. Let $O$ be an arbitrary dataset of $\eta n$ examples in $\mathcal{X} \times \{-1, 1\}$ and let $\mathbf{w}$ be an optimum to $(SLM_{\ell, \gamma})$ on $I \cup O$, for an L-admissible loss $\ell$ and for $\gamma \leq \gamma^\star$. Let $(\mathbf{z}, y)$ satisfy $y\mathbf{z}^\top \mathbf{w}^\star \geq 1$ and suppose that the pancake $P_\mathbf{w}^\tau(\mathbf{z}, y)$ is $\rho$-dense with respect to $I$ for $\tau \leq \gamma/2$. If $(1 - \eta)\rho \sqrt{(\gamma^\star)^2 - \tau^2} n > L \cdot$ LinSumNorm$(O)$, then $\mathbf{z}$ is not misclassified by $\mathbf{w}$.*

*Proof.* Let $\mathbf{w}$ be an optimum of $(SLM_{\ell, \gamma})$ and let $\mathbf{v}$ denote $\frac{\mathbf{w}}{\|\mathbf{w}\|}$. Let $\mathbf{v}^\star$ denote $\frac{\mathbf{w}^\star}{\|\mathbf{w}^\star\|}$. First suppose that $\mathbf{v}^\star$ has a non-zero component orthogonal to $\mathbf{v}$; let $\mathbf{v}'$ be a unit vector in the direction of this component. I.e. $\mathbf{v}' = \frac{\mathbf{v}^\star - (\mathbf{v}^\top \mathbf{v}^\star)\mathbf{v}}{\|\mathbf{v}^\star - (\mathbf{v}^\top \mathbf{v}^\star)\mathbf{v}\|}$.

Suppose that an example $(\mathbf{z}, y)$ satisfying the margin condition $y\mathbf{z}^\top \mathbf{w}^\star \geq 1$ is misclassified by $\mathbf{w}$, i.e $y\mathbf{z}^\top \mathbf{w} \leq 0$. Let $I_1 = \{(\mathbf{x}_i, y_i) \in I : y\mathbf{v}^\top \mathbf{z} - \tau \leq y_i \mathbf{v}^\top \mathbf{x}_i \leq y\mathbf{v}^\top \mathbf{z} + \tau\}$ denote the set of examples in $P_\mathbf{v}^\tau(\mathbf{z}, y) \cap I$. Let $I_2$ denote $I \setminus I_1$. Then we write:

$$\nabla_\mathbf{w} \ell = \sum_{(\mathbf{x}_i, y_i) \in I_1} \nabla_\mathbf{w} \ell(\mathbf{w}; \mathbf{x}_i, y_i) + \sum_{(\mathbf{x}_i, y_i) \in I_2} \nabla_\mathbf{w} \ell(\mathbf{w}; \mathbf{x}_i, y_i) + \sum_{(\mathbf{x}_i, y_i) \in O} \nabla_\mathbf{w} \ell(\mathbf{w}; \mathbf{x}_i, y_i). \tag{1}$$

We defer the proofs of the following two lemmas to Appendix A.

**Lemma 11.** *Let $(\mathbf{x}_i, y_i) \in I_1$ where $I_1, \mathbf{w}, \mathbf{v}$ are defined as above. Then*

1. $y_i \mathbf{v}^\top \mathbf{x}_i \leq \tau \leq \gamma/2$, *so that $f'(y_i \mathbf{w}^\top \mathbf{x}_i) \leq -1$.*

2. $y_i \mathbf{x}_i^\top \mathbf{v}' \geq \sqrt{(\gamma^\star)^2 - \tau^2}$.

**Lemma 12.** *For any set $O \subseteq \mathcal{X} \times \{-1, +1\}$, and any $\mathbf{w}$,*

$$\|\sum_{(\mathbf{x}_i, y_i) \in O} \nabla_\mathbf{w} \ell(\mathbf{w}; \mathbf{x}_i, y_i)\| \leq L \cdot \text{LinSumNorm}(O).$$

The rest of the proof argues that contribution from the misclassified points in $I_1$ to the gradient in (1) cannot be compensated for by the small number of points in $O$. We consider two separate cases, depending on whether or not the constraint $\|\mathbf{w}\| \leq \frac{1}{\gamma}$ in the convex program ($SLM_{\ell,\gamma}$) is tight for $\mathbf{w}$.

**Case 1:** $\|\mathbf{w}\| < \frac{1}{\gamma}$: In this case, the optimum to the constrained program is in the interior of the constraint set, so that the gradient of the objective function at the optimum $\mathbf{w}$ is zero. In particular, this implies that:

$$(\nabla_{\mathbf{w}}\ell)^{\top}\mathbf{v}^{\star} = 0. \tag{2}$$

We look at the three terms in (1). For the first term, Lemma 11 implies that each point in $I_1$ contributes a non-trivial amount to $(\nabla_{\mathbf{w}}\ell)^{\top}\mathbf{v}^{\star}$. Indeed for every $(\mathbf{x}_i, y_i) \in I_1, f'(y_i\mathbf{w}^{\top}\mathbf{x}_i) \leq -1$ and $y_i\mathbf{x}_i^{\top}\mathbf{v}^{\star} \geq \gamma^{\star}$ so that

$$\sum_{(\mathbf{x}_i,y_i)\in I_1} (\nabla_{\mathbf{w}}\ell(\mathbf{w}; \mathbf{x}_i, y_i))^{\top}\mathbf{v}^{\star} \leq -\gamma^{\star}|I_1| \leq -\gamma^{\star}(1-\eta)\rho n. \tag{3}$$

Moreover, for any point $(\mathbf{x}_i, y_i) \in I_2$, the product $y_i\mathbf{x}_i^{\top}\mathbf{v}^{\star} \geq \gamma^{\star} > 0$. Since $f'(\cdot) \leq 0$, it follows that

$$\sum_{(\mathbf{x}_i,y_i)\in I_2} (\nabla_{\mathbf{w}}\ell(\mathbf{w}; \mathbf{x}_i, y_i))^{\top}\mathbf{v}^{\star} \leq 0. \tag{4}$$

Finally, since $\mathbf{v}^{\star}$ is a unit vector, Lemma 12 and Cauchy-Schwartz imply that

$$\left(\sum_{(\mathbf{x}_i,y_i)\in O} \nabla_{\mathbf{w}}\ell(\mathbf{w}; \mathbf{x}_i, y_i)\right)^{\top} \mathbf{v}^{\star} \leq L \cdot \text{LinSumNorm}(O). \tag{5}$$

Adding together (3), (4) and (5), we get an upper bound on $(\nabla_{\mathbf{w}}\ell)^{\top}\mathbf{v}^{\star}$. Under the assumptions on the parameters, this upper bound is negative, contradicting (2).

**Case 2:** $\|\mathbf{w}\| = \frac{1}{\gamma}$: By the optimality of $\mathbf{w}$, KKT conditions imply that the gradient of the loss must be in the span of the gradients of the constraints. In our case, this implies that the gradient is a scalar multiple of $\mathbf{v}$ and thus orthogonal to $\mathbf{v}'$:

$$(\nabla_{\mathbf{w}}\ell)^{\top}\mathbf{v}' = 0. \tag{6}$$

We will once again look at the three terms in (1). For the first term, Lemma 11(2) now implies that

$$\sum_{(\mathbf{x}_i,y_i)\in I_1} (\nabla_{\mathbf{w}}\ell(\mathbf{w}; \mathbf{x}_i, y_i))^{\top}\mathbf{v}' \leq -\sqrt{(\gamma^{\star})^2 - \tau^2}|I_1| \leq -\sqrt{(\gamma^{\star})^2 - \tau^2}(1-\eta)\rho n \tag{7}$$

The argument for the second term is now somewhat more complicated. Let $(\mathbf{x}_i, y_i) \in I_2$. If $y_i\mathbf{x}_i^{\top}\mathbf{v}' \geq 0$, then clearly $(\nabla_{\mathbf{w}}\ell(\mathbf{w}; \mathbf{x}_i, y_i))^{\top}\mathbf{v}' \leq 0$. On the other hand, if $y_i\mathbf{x}_i^{\top}\mathbf{v}' < 0$, then by definition of $\mathbf{v}'$,

$$0 > y_i\mathbf{x}_i^{\top}(\mathbf{v}^{\star} - (\mathbf{v}^{\top}\mathbf{v}^{\star})\mathbf{v})$$
$$= y_i\mathbf{x}_i^{\top}\mathbf{v}^{\star} - (\mathbf{v}^{\top}\mathbf{v}^{\star})y_i\mathbf{x}_i^{\top}\mathbf{v},$$

which, coupled with $\mathbf{v}^{\top}\mathbf{v}^{\star} < 1$ implies that

$$y_i\mathbf{x}_i^{\top}\mathbf{v} \geq y_i\mathbf{x}_i^{\top}\mathbf{v}^{\star}/(\mathbf{v}^{\top}\mathbf{v}^{\star}) > \gamma^{\star} \geq \gamma.$$

Thus $f'(y_i\mathbf{x}_i^{\top}\mathbf{v}) = 0$, so that $(\nabla_{\mathbf{w}}\ell(\mathbf{w}; \mathbf{x}_i, y_i))^{\top}\mathbf{v}' = 0$. It follows that

$$\sum_{(\mathbf{x}_i,y_i)\in I_2} (\nabla_{\mathbf{w}}\ell(\mathbf{w}; \mathbf{x}_i, y_i))^{\top}\mathbf{v}' \leq 0. \tag{8}$$

Finally, as before Lemma 12 implies that

$$\left(\sum_{(\mathbf{x}_i,y_i)\in O} \nabla_{\mathbf{w}}\ell(\mathbf{w}; \mathbf{x}_i, y_i)\right)^{\top} \mathbf{v}' \leq L \cdot \text{LinSumNorm}(O). \tag{9}$$

Adding together (7), (8) and (9), we get an upper bound on $(\nabla_{\mathbf{w}}\ell)^\top \mathbf{v}'$. Under the assumptions on the parameters, this upper bound is negative, which contradicts (6).

Finally, we deal with the easy case where $\mathbf{v}'$ does not exist, i.e. $\mathbf{v}^\star$ and $\mathbf{v}$ are collinear.

**Case 3: $\mathbf{v} = \alpha\mathbf{v}^\star$.** Since $\mathbf{v}$ and $\mathbf{v}^\star$ are unit vectors, $\mathbf{v} \in \{\mathbf{v}^\star, -\mathbf{v}^\star\}$. If $\mathbf{v} = \mathbf{v}^\star$, $\mathbf{w}$ defines the same classifier as $\mathbf{w}^\star$ and we are done. If $\mathbf{v} = -\mathbf{v}^\star$, we will argue that $(\nabla_{\mathbf{w}}\ell)^\top \mathbf{v}^\star > 0$, which contradicts the optimality of $\mathbf{w}$. In this case every inlier is misclassified by $\mathbf{w}$, and contributes at least $\gamma$ to the $(\nabla_{\mathbf{w}}\ell)^\top \mathbf{v}^\star$. On the other hand, the contribution from the outliers is bounded in norm by $L \cdot \mathrm{LinSumNorm}(O)$ by Lemma 12. By assumption, the contribution from the inliers is larger than that from the outliers, leading to a contradiction. The claim follows. $\qquad\square$

**Remark 1.** *The proof does not quite require that we reach an optimum, since the norm of the gradient can be lower bounded if we allow a small slack in the condition $(1-\eta)\rho\sqrt{(\gamma^\star)^2 - \tau^2}n > L \cdot \mathrm{LinSumNorm}(O)$. Thus the lemma holds not just for the optimizer to the ERM, but to any point $\mathbf{w}$ with a suitably small gradient. Points satisfying such bounded gradient condition would result, e.g. by running a stochastic gradient descent algorithm on a smooth admissible loss.*

**Remark 2.** *The theorem allows for $f$ itself to be non-convex as long as it is admissible, when $\mathbf{w}$ is an approximate first-order critical point.*

**Remark 3.** *The proof only needed the pancake condition to hold in the direction of the empirical solution $\mathbf{w}$. This can be easier to verify on a clean validation set, as compared to testing the dense pancake condition along all directions.*

The lemma immediately implies the following theorem.

**Theorem 13.** *Let $\mu$ be a $\gamma^\star$-margin separable distribution and let $I$ be a sample of $(1-\eta)n$ examples drawn from $\mu$. Let $O$ be an arbitrary dataset of $\eta n$ examples in $\mathcal{X} \times \{-1, 1\}$ and let $D = I \cup O$. Let $\mathbf{w}$ be an optimum to $(SLM_{\ell,\gamma})$ on $D$, for an L-admissible loss $\ell$ and for $\gamma \leq \gamma^\star$. If $(D, \mu)$ satisfies the $(\tau, \rho, \beta)$-dense pancakes condition for $\tau \leq \gamma^\star/2$ and $(1-\eta)\rho\sqrt{(\gamma^\star)^2 - \tau^2}n > L \cdot \mathrm{HerSumNorm}(O)$, then $\mathbf{w}$ has accuracy at least $(1-\beta)$ on $\mu$.*

## 4 Bounding the Sum Norm

We can prove that for any isotropic distribution over $\mathcal{X}$, the HerSumNorm for any subset is small. The proof uses some results in random matrix theory due to Adamczak et al. [2010], and is deferred to Appendix D.

**Theorem 14.** *Let $\{\mathcal{D}_j\}$ be logconcave distributions on $\mathbb{R}^d$ such that $\|\mathbb{E}_{\mathbf{x}\sim\mathcal{D}_j}[\mathbf{x}]\| \leq \mu$ and $\|\mathbb{E}_{\mathbf{x}\sim\mathcal{D}_j}[\mathbf{x}\mathbf{x}^\top]\| \leq \frac{1}{d}$. Let $\mathcal{D}$ be a mixture of $\mathcal{D}_j$'s with arbitrary weights. Let $n \leq \exp(\sqrt{d})$ and let $X_1, \ldots, X_n$ be samples from $\mathcal{D}$. Then except with probability $\exp(-c\sqrt{d})$, every subset $O \subseteq \{X_1, \ldots, X_n\}, |O| \leq m$ satisfies:*

$$\mathrm{LinSumNorm}(O) \leq C(\sqrt{m} + m\mu + \frac{m\log\frac{2n}{m}}{\sqrt{d}}).$$

We note that the homogenization transform that replaces the classifier $\mathrm{sgn}(\mathbf{w}^\top\mathbf{z} + b)$ in $\mathbb{R}^d$ by the homogenous classifier $\mathrm{sgn}((\mathbf{w}, b/\mu)^\top(\mathbf{z}, \mu))$ in $\mathbb{R}^{d+1}$ changes the sum norm of $O$ by an additive $m\mu$, and the norm of $\mathbf{w}$ by a constant factor as long as $|b| \leq \mu/\gamma$. Values of $|b|$ larger than $\mu/\gamma$ are uninteresting, since most of the mass of a distributions such as above would then lie on one side of the hyperplane. Thus our assumption on the classifiers being homogenous is essentially without loss of generality.

## 5 Adversarial Label Noise

In this section, we use the results from the last several sections to establish noise resistance of surrogate loss minimization (SLM) in probabilistic data models. We focus on Adversarial Label Noise here, and look at some other noise models in the Appendix.

In the adversarial label noise model of [Haussler, 1992, Kearns et al., 1994], the adversary can flip the labels on an arbitrary $\eta$ fraction of the examples, but cannot change the points themselves.

**Theorem 15** (Robustness under Adversarial Label noise)**.** *Let* $\{\mathcal{D}_j\}$ *be a set of* $O(1)$ *logconcave distributions on* $\mathcal{X} = \{\mathbf{x} \in \mathbb{R}^d : \|\mathbf{x}\|_2 \leq 1\}$ *such that* $\|\mathbb{E}_{\mathbf{x}\sim\mathcal{D}_j}[\mathbf{x}]\| \leq \mu$ *and* $\|\mathbb{E}_{\mathbf{x}\sim\mathcal{D}_j}[\mathbf{x}\mathbf{x}^\top]\| \preceq \mathbb{I}/d$. *Let* $\mathcal{D}$ *be a uniform mixture of* $\mathcal{D}_j$'s. *Further suppose that this distribution is* $\gamma$-*margin separable for* $\gamma \geq C \log \frac{1}{\beta}/\sqrt{d}$. *Then for adversarial label error rate* $\eta = O(\min(\frac{1}{3}, \frac{\gamma}{2\mu}, \frac{\gamma\sqrt{d}}{\log(\gamma\sqrt{d})}))$, *an SLM learnt on* $n$ *samples from* $\mathcal{D}$ *has accuracy* $1 - \beta$ *on the inlier distribution as long as* $n \in (\Omega(d \ln \frac{1}{\gamma} + \ln \frac{1}{\beta}), exp(O(\sqrt{d})))$.

*Proof.* Under the assumptions, the LinSumNorm($O$) is bounded by $O(\sqrt{m}+m\mu+m\frac{\log 2n/m}{\sqrt{d}})$. Moreover, the distribution satisfies the $(\gamma/2, \Omega(1), \beta)$-dense pancakes condition. Applying Theorem 13, the claim follows. $\square$

Note that a set of random points from $\mathcal{N}(0, \frac{1}{d}\mathbb{I})$ will have margin approximately $\Omega_\beta(\frac{1}{\sqrt{d}})$ for any classifier, for all but a $\beta$ fraction of the points. We call this margin "trivial". The above result says that if the margin is a constant factor better than trivial, then under the other assumptions, a *constant* rate of adversarial errors can be tolerated.

The Massart Noise model [Boucheron et al., 2005] is a special case of adversarial label noise, where the adversary can only specify a flipping probability $\eta(\mathbf{x})$ for each example, subject to $\eta(\mathbf{x}) \leq \eta$ for all $\mathbf{x}$. Thus the bounds for the adversarial label noise model extend to this setting.

## Broader Impact Statement

This work explains the robustness of a commonly used learning algorithm to errors in the training data, and is a step in the broader research direction of making machine learning robust to outliers. Over the longer term, this research direction will allow training on larger noisier data sets that may be available to some entities, a consequence it shares with a large majority of research in machine learning. This may carry some risks such as increasing inequities in access to useful training data and challenges in inspecting these datasets for bugs and biases. Nevertheless, given the evidence of more data being helpful along many different axes, and machine learning being useful at large, we believe research in this area is a net positive.

## 6 Acknowledgements

I would like to thank Yoram Singer, Ludwig Schmidt, Phil Long, Tomer Koren and Satyen Kale for numerous useful discussions on this work. I would also like to thank the anonymous referees for their feedback.

## Footnotes

*Work performed while at Google Brain.

[2]In this introduction, we use the term outliers to refer to the corrupted training data.

[3]Note that general linear classifiers may have a bias term. One can easily absorb this bias term by adding another dimension. This transformation to *homogenous* linear classifies is standard (e.g. Shalev-Shwartz and Ben-David [2014, Ch. 9]) and helps simplify notation. The deterministic conditions we require are robust to this transformation as we discuss in Appendix B and Section 4

[4]when $f$ is not convex, this may be a "local" subderivative.

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
