[Supplementary Material]

# Supplementary Material for "On the Error Resistance of Hinge Loss Minimization"

## A  Missing Proofs from Section 3

*Proof of Lemma 11.* Recall that $y\mathbf{z}^\top \mathbf{v} \leq 0$ and that $(\mathbf{x}_i, y_i)$ is in $P^\tau_\mathbf{v}(\mathbf{z}, y)$ so that $y_i \mathbf{v}^\top \mathbf{x}_i \leq \tau$; by the assumption on $\tau$, this is at most $\gamma/2$. The fact that $\|\mathbf{w}\| \leq \frac{1}{\gamma}$ then implies the first claim.

Let $\mathbf{v}^\top \mathbf{v}^\star = \alpha$. Note that

$$y_i \mathbf{x}_i^\top (\mathbf{v}^\star - (\mathbf{v}^\top \mathbf{v}^\star)\mathbf{v}) \geq \gamma^\star - (\mathbf{v}^\top \mathbf{v}^\star)\tau$$
$$= \gamma^\star - \alpha\tau,$$

whereas

$$\|\mathbf{v}^\star - (\mathbf{v}^\top \mathbf{v}^\star)\mathbf{v}\|^2 = 1 + \alpha^2 - 2\alpha^2$$
$$= 1 - \alpha^2.$$

Thus

$$y_i \mathbf{x}_i^\top \mathbf{v}' \geq \frac{\gamma^\star - \alpha\tau}{\sqrt{1 - \alpha^2}}.$$

Setting the derivative with respect to $\alpha$ to zero, we can verify that this expression is minimized when $\alpha = \frac{\tau}{\gamma^\star}$. Plugging in this value immediately yields the result. □

*Proof of Lemma 12.* Since $|f'(\cdot)| \leq L$, it follows that

$$\| \sum_{(\mathbf{x}_i, y_i) \in O} \nabla_\mathbf{w} \ell(\mathbf{w}; \mathbf{x}_i, y_i) \| = \| \sum_{(\mathbf{x}_i, y_i) \in O} f'(y_i \mathbf{x}_i^\top \mathbf{w}) \cdot y_i \mathbf{x}_i \|$$

$$= L \cdot \| \sum_{(\mathbf{x}_i, y_i) \in O} \left( \frac{y_i f'(y_i \mathbf{x}_i^\top \mathbf{w})}{L} \right) \cdot \mathbf{x}_i \|$$

$$\leq L \cdot \text{LinSumNorm}(O).$$

□

## B  Proving the Density Condition

In this section, we develop tools to prove the pancake condition on distributions. We first show that for a large class of distributions, the pancake condition is satisfied for appropriate parameters.

**Theorem 16.** *Let $\mu$ be the isotropic Gaussian distribution $\mathcal{N}(0, \sigma^2 \mathbb{I})$ and let $h : \mathbb{R}^d \to \mathbb{R}$ be an arbitrary linear classifier. Let $\mathcal{D}$ be the distribution $(\mathbf{x}, h(\mathbf{x})) : \mathbf{x} \sim \mu$ and let $(\mathbf{z}, y)$ be a sample from this distribution. Then for any $\beta > 0$, $\mathcal{D}$ satisfies the $(2\sigma\sqrt{2 \ln \frac{1}{\beta}}, 1 - \beta, \beta)$-dense pancake condition.*

*Proof.* Fix a unit vector $\mathbf{w}$. Then the distribution $\mathbf{x}^\top \mathbf{w}$ is a $\mathcal{N}(0, \sigma^2)$. All but a $(1 - \beta)$ fraction of the mass of the Gaussian is contained in $[-\sigma\sqrt{2 \ln \frac{1}{\beta}}, \sigma\sqrt{2 \ln \frac{1}{\beta}}]$. Thus for $\tau = 2\sigma\sqrt{2 \ln \frac{1}{\beta}}$, this probability $\rho \geq 1 - \beta$. □

The theorem extends to distributions $\mu$ more general than Gaussians. Any distribution $\mu$ satisfying the Herbst condition $\mathbb{E}_{\mathbf{z} \sim \mu}[\exp(\varepsilon\|\mathbf{z}\|_2^2)] < \infty$ for some $\varepsilon > 0$ satisfies an analog of Theorem 16 with the constant in front of $\tau$ depending on $\varepsilon$. More generally, whenever $\mu$ is strongly log concave, i.e. has density $\exp(-V(\mathbf{x}))$ at $\mathbf{x}$, where $V$ is a strongly convex function satisfying $Hes(V) \succeq c\mathbb{I}$ (see Bobkov [1999, Thm. 1.3]), an analogous theorem holds with constants depending on $c$. Moreover, any isotropic logconcave distribution satisfies a weaker form of this result, with the $\sqrt{\log \beta^{-1}}$ being replaced by a $\log \beta^{-1}$.

The dense pancakes condition is invariant to rotation and translation and behaves nicely under affine transformations.

**Theorem 17.** *Let $\mu$ satisfy the $(\tau, \rho, \beta)$-dense pancakes condition for $\tau > 0, \rho, \beta \in (0, 1)$. Then for an affine map $A : \mathbb{R}^d \to \mathbb{R}^{d'}$ with lipschitz constant c, the push-forward distribution satisfies the $(c\tau, \rho, \beta)$-dense pancakes condition.*

Note that in particular, this theorem means that the transformation $\mathbf{x} \to (\mathbf{x}, 1)$ used to make the classifier homogeneous in Section 2 preserves the dense pancake condition without any change in parameters. It is also easy to see that this condition is robust to small changes in the distribution.

**Theorem 18.** *Let $\mu$ satisfy the $(\tau, \rho, \beta)$-dense pancakes condition for $\tau > 0, \rho, \beta \in (0, 1)$ and suppose that $\mu'$ is at statistical distance $\beta'$ from $\mu$ for $\beta' \in (0, 1)$. Then $\mu'$ satisfies the $(\tau, \rho - \beta', \beta + \beta')$-dense pancakes condition. If $\nu$ is at Wasserstein distance (aka earthmover distance) $\Delta$ from $\mu$, then for any $\beta' \in (0, 1)$, $\nu$ satisfies the $(\tau + \frac{2\Delta'}{\beta}, \rho - \beta', \beta + \beta')$-dense pancakes condition.*

Moreover, mixtures of such distributions continue to satisfy this condition.

**Theorem 19.** *Suppose that $\mu_1, \ldots, \mu_k$ are such that for each i, $\mu_i$ satisfies the $(\tau, \rho, \beta)$-dense pancake condition for $\tau > 0, \rho, \beta \in (0, 1)$. Let $\bar{\mu}$ denote the mixture distribution defined by sampling from a $\mu_i$ with i chosen u.a.r. from $[k]$. Then $\bar{\mu}$ satisfies the $(\tau, \frac{\rho}{k}, \beta)$-dense pancakes condition.*

This allows us to assert the dense pancakes condition for many distributions $\mathcal{D}$ of interest. E.g. suppose that the class conditional distributions are Gaussians with variance $1/\sqrt{d}$ in each direction. Then the distributions satisfies the $(c\sqrt{\frac{\log \beta^{-1}}{d}}, \frac{1-\beta}{2}, \beta)$-dense pancakes condition.

## C   Pancakes Condition: Distributions to Empirical

In this section, we show that if a distribution satisfies the dense pancakes condition, then so does an empirical sample from it.

**Theorem 20.** *Suppose that $\mu$ satisfies the $(\tau, \rho, \beta)$ dense pancakes condition. Let D be a sample of n examples chosen i.i.d. from $\mu$. Then $(D, \mu)$ satisfies the $(\tau + \tau', \rho/2, \beta + \beta')$ dense pancakes condition with probability $(1 - \exp(-d))$ as long as*

$$ n \geq \frac{8}{\rho} \cdot \left( d(\log(1 + \frac{2}{\tau'})) + \log \frac{1}{\beta'} \right) $$

*Proof.* The proof follows a standard recipe of using measure concentration results along with a union bound over a net. We say that a sample $D$ is $(\tau, \rho, \beta)$-good for $\mathbf{w}$ if $\mu(\{(\mathbf{z}, y) : P_{\mathbf{w}}^{\tau}(\mathbf{z}, y) \text{ is not } \rho\text{-dense w.r.t. } D\}) \leq \beta$. Our goal is to show that with high probability over the choice of $D$, it is the case that for every unit vector $\mathbf{w}$, the sample $D$ is $(\tau + \tau', \rho/2, \beta + \beta')$-good for $\mathbf{w}$.

First fix a unit vector $\mathbf{w}$. Let $S = \{(\mathbf{z}, y) : P_{\mathbf{w}}^{\tau}(\mathbf{z}, y) \text{ is not } \rho\text{-dense w.r.t. } \mu\}$. Since $\mu$ satisfies the $(\tau, \rho, \beta)$-dense pancakes condition, $\mu(S) \leq \beta$. For any $(\mathbf{z}, y) \in S^c$, we have that $\mu(P_{\mathbf{w}}^{\tau}(\mathbf{z}, y)) \geq \rho$. Since $D$ is formed by taking $n$ i.i.d. samples from $\mu$, Chernoff bounds tell us that

$$ \Pr_D \left[ \sum_{(\mathbf{x}_i, y_i) \in D} \mathbb{1}((\mathbf{x}_i, y_i) \in P_{\mathbf{w}}^{\tau}(\mathbf{z}, y)) \leq \rho n/2 \right] \leq \exp(-\rho n/8). $$

In other words, for any $(\mathbf{z}, y) \in S^c$,

$$ \Pr_D \left[ \mu_D(P_{\mathbf{w}}^{\tau}(\mathbf{z}, y)) \leq \rho/2 \right] \leq \exp(-\rho n/8). $$

It follows that

$$ \mathbb{E}_D \left[ \mu \left( \mathbb{1}((\mathbf{x}_i, y_i) \in S^c \ \wedge \ \mu_D(P_{\mathbf{w}}^{\tau}(\mathbf{z}, y)) \leq \rho/2) \right) \right] \leq \exp(-\rho n/8), $$

so that by Markov's inequality, for any $\beta' > 0$,

$$ \Pr_D[\mu(\mathbb{1}((\mathbf{x}_i, y_i) \in S^c \ \wedge \ \mu_D(P_{\mathbf{w}}^{\tau}(\mathbf{z}, y)) \leq \rho/2)) \geq \beta'] \leq \beta'^{-1} \cdot \exp(-\rho n/8). $$

This coupled with the fact that $\mu(S) \leq \beta$ implies that

$$ \Pr_D[\mu(\mathbb{1}(\mu_D(P_{\mathbf{w}}^{\tau}(\mathbf{z}, y)) \leq \rho/2)) \geq \beta + \beta'] \leq \beta'^{-1} \cdot \exp(-\rho n/8). $$

This implies that the sample $D$ is $(\tau, \rho/2, \beta+\beta')$-good with respect to a fixed $\mathbf{w}$, except with probability $\beta'^{-1} \cdot \exp(-\rho n/8)$.

Next, note that if $\|\mathbf{w} - \mathbf{w}'\| \le \tau'$, then $P_{\mathbf{w}}^{\tau}(\mathbf{z}, y) \subseteq P_{\mathbf{w}'}^{\tau+\tau'}(\mathbf{z}, y)$. Thus it suffices to do a union bound over a $\tau'$-net of the unit ball. Since one can find such a net (see e.g. Vershynin [2010, Lemma 5.2]) of size $\exp(d \log(1 + \frac{2}{\tau'}))$, it follows that

$$\Pr_D[D \text{ is not } (\tau + \tau', \rho/2, \beta + \beta')\text{-good for all } \mathbf{w}] \le \exp(d \log(1 + \frac{2}{\tau}') + \log\frac{1}{\beta'} - \frac{\rho n}{8}).$$

Plugging in the bound on $n$, the claim follows. $\qquad\square$

We remark that the linear dependence on $d$ can be replaced by a $\frac{1}{\gamma^2}$ by random projections. Whether or not this additional projection step is needed is an interesting open question.

## D  Bounding the Sum Norm

We will use a result from Adamczak et al. [2010, Theorem 3.6], who study the norm of restrictions of matrices consisting of samples from a log concave isotropic distribution. Let $X_1, \ldots, X_n$ be samples from an isotropic logconcave distribution in $\mathbb{R}^d$ and for $S \subseteq [n]$, let $A_S$ be the $d \times |S|$ matrix consisting of $X_i, i \in S$ as columns. The following is a restatement of their result:

**Theorem 21** (Adamczak et al. [2010]). *Let $d \ge 1$ and $n \le \exp(\sqrt{d})$ be integers. Let $X_1, \ldots, X_n$ be samples from an isotropic logconcave distribution in $\mathbb{R}^d$, and let $A_S$ be defined as above. Then there are absolute constants $C, c$ such that for any $K \ge 1$,*

$$\Pr[\exists S \subseteq [n] : \|A_S\| \ge CK(\sqrt{d} + \sqrt{|S|}\log\frac{2n}{|S|})] \le \exp(-cK\sqrt{d}).$$

*Here $\|A\| = \sup_{z \in S^{d-1}} |Az|$ denotes the norm of $A$.*

**Corollary 22.** *Let $d \ge 1$ and $n \le \exp(\sqrt{d})$ be integers. Let $X_1, \ldots, X_n$ be samples from an logconcave distribution with covariance $\frac{1}{d}\mathbb{I}$ in $\mathbb{R}^d$. Then except with probability $\exp(-c\sqrt{d})$, every subset $O \subseteq \{X_1, \ldots, X_n\}, |O| \le m$ satisfies*

$$\text{LinSumNorm}(O) \le C(\sqrt{m} + \frac{m\log\frac{2n}{m}}{\sqrt{d}}).$$

*Proof.* By scaling, $\sqrt{d}X_i$'s are samples from an isotropic logconcave distribution. By Lemma 9 above, it suffices to bound the HerSumNorm of $O$. Applying Theorem 21 and noting the scaling by up to $\sqrt{m}$ to bring an indicator vector on $O$ to $S^{m-1}$, the claimed bound follows. $\qquad\square$

The assumption on $n$ is only needed to ensure that the norm of the largest sample is bounded. An analog of this theorem without this restriction can be proved, where there is an additional term corresponding to $\max_i \|X_i\|$. In particular, if our distributions are supported on bounded norm vectors, this restriction on $n$ is unnecessary.

We can now extend this bound to mixture distributions and prove Theorem 14.

*Proof of Theorem 14.* By paying at most $m\mu$ in the total sum norm, we can shift all the means to the origin. Now the mixture distribution is itself logconcave and has covariance matrix $\Sigma \preceq \frac{1}{d}\mathbb{I}$. The covariance can be made to equal $\frac{1}{d}\mathbb{I}$ by adding additional noise and it is easy to show that the additional noise only increases the sum norm. $\qquad\square$

## E  Linear versus Hereditary Sum Norm

We restate and prove Lemma 9

**Lemma 23.** *Let $D \subseteq \mathcal{X} \times \{-1, +1\}$ Then* $\text{HerSumNorm}(D) = \text{LinSumNorm}(D)$.

*Proof.* We rewrite the definitions as

$$\text{LinSumNorm}(D) = \sup_{a_1,\ldots,a_{|D|}:0\le a_i\le 1} \Big\| \sum_{(\mathbf{x}_i,y_i)\in D} a_i \mathbf{x}_i \Big\|$$

$$\text{HerSumNorm}(D) = \sup_{a_1,\ldots,a_{|D|}:a_i\in\{0,1\}} \Big\| \sum_{(\mathbf{x}_i,y_i)\in D} a_i \mathbf{x}_i \Big\|.$$

This rephrasing makes it immediate the $\text{LinSumNorm}(D) \ge \text{HerSumNorm}(D)$.

For the other direction, let $a_1,\ldots,a_{|D|}$ be scalars in $[0,1]$ that achieve[5] the sup in the definition of $\text{LinSumNorm}(D)$. Consider the following randomized rounding:

$$A_i = \begin{cases} 1 & \text{w.p. } a_i \\ 0 & \text{otherwise} \end{cases}$$

We claim that $\mathbb{E}[\|\sum_{(\mathbf{x}_i,y_i)\in D} A_i\mathbf{x}_i\|_2^2] \ge \|\sum_{(\mathbf{x}_i,y_i)\in D} a_i\mathbf{x}_i\|_2^2$. Indeed it suffices to prove this for a single co-ordinate $j$. Clearly $\mathbb{E}[\sum_{(\mathbf{x}_i,y_i)\in D} A_i\mathbf{x}_{ij}] = \sum_{(\mathbf{x}_i,y_i)\in D} a_i\mathbf{x}_{ij}$. By Jensen's inequality, $\mathbb{E}[(\sum_{(\mathbf{x}_i,y_i)\in D} A_i\mathbf{x}_{ij})^2] \ge (\sum_{(\mathbf{x}_i,y_i)\in D} a_i\mathbf{x}_{ij})^2$. The claim follows. $\qquad\square$

# F   Applications to Various Noise Models

In this section, we use the results from the last several sections to establish noise resistance of surrogate loss minimization (SLM) in probabilistic data models. We look at some additional noise models that differ in the constraints placed on the outliers. Some of these results only serve primarily to demonstrate the unified framework and reprove known bounds for other algorithms.

**Malicious Noise Model**

In the malicious noise model [Valiant, 1985, Kearns and Li, 1988], the outliers are arbitrary. In a variant of this model known as the nasty noise model [Bshouty et al., 2002], the outliers can depend on the inliers samples, and not just on the inlier distribution. The following result captures the robustness of SLM in this setting.

**Theorem 24** (Robustness under nasty/malicious noise). *Suppose that the inlier distribution $\mathcal{D}$ has a margin $\gamma$ and satisfies the $(\gamma/2, \rho, \beta)$-dense pancake condition. Then for malicious error rate $\eta = O(\gamma\rho)$, the SLM has accuracy $1 - \beta$ on the inlier distribution.*

*Proof.* An adversarial set of outliers satisfies the property that $\text{LinSumNorm}(O) \le |O| \le \eta n$. The claim then follows from Theorem 13. $\qquad\square$

**Random Classification Noise Model** This model was introduced by [Angluin and Laird, 1988], where a random $\eta$ fraction of the examples have their labels flipped. In this case, the outliers are simply $\eta n$ random samples from the same distribution. The result can be slightly improved for this case.

**Theorem 25** (Robustness under Random Label noise). *Let $\{\mathcal{D}_j\}$ be a set of $O(1)$ logconcave distributions on $\mathbb{R}^d$ such that $\|\mathbb{E}_{\mathbf{x}\sim\mathcal{D}_j}[\mathbf{x}]\| \le \mu$ and $\|\mathbb{E}_{\mathbf{x}\sim\mathcal{D}_j}[\mathbf{x}\mathbf{x}^\top]\| \le 1/d$. Let $\mathcal{D}$ be a uniform mixture of $\mathcal{D}_j$'s. Further suppose that this distribution is $\gamma$-margin separable for $\gamma \ge C\log\frac{1}{\beta}/\sqrt{d}$. Then for adversarial label error rate $\eta = O(\min(\frac{1}{3}, \frac{\gamma}{2\mu}, \gamma\sqrt{d}))$, the SLM has accuracy $1 - \beta$ on the inlier distribution.*

*Proof.* Under the assumptions, we can now apply Theorem 14 with $m = n$, since the adversary cannot select the points to corrupt any more. In this case, the $\text{LinSumNorm}(O)$ is bounded by $O(\sqrt{m} + m\mu + m/\sqrt{d})$. Moreover, the distribution satisfies the $(\gamma/2, \Omega(1), \beta)$-dense pancakes condition. Applying Theorem 13, the claim follows. $\qquad\square$

## Footnotes

[5]Since $[0,1]^{|D|}$ is compact, the sup is indeed achieved.