[Reviews · NeurIPS 2020]

Review 1

Summary and Contributions: This paper studies when hinge loss for the purpose of learning a linear classifier is resistant to malicious noise. Since this is a provably hard problem, it is common to make assumptions on the data generating distribution, e.g., uniform or log-concave. This paper works with a different assumption regarding the margin called the Dense Pancakes condition. Roughly speaking this condition means that when xy are projected onto a unit vector most point are close to many other points. The paper shows that when these conditions hold and the original distribution is in a bounded ball, then hinge loss gives a good error bound. UPDATE: I have read the response and I'm happy to keep my original score.

Strengths: Overall, I like the paper. Hinge loss minimization is used as a subroutine or by itself for many application and understanding when it works is a valuable effort. It’s interesting to me that in many other applications hinge loss has to be combined with other techniques, or be iteratively applied in different conditional distributions, for it to have good performance. Whereas, this work shows that hinge loss can be pretty effective by itself.

Weaknesses: This is not a weakness per se, but I did find the following characterization to be misguiding: “Additionally all existing works in this line of work aim to learn a classifier that has error rate (on  inliers and outliers) close to the corruption rate eta”. This is not true. In fact, the first works that showed halfspaces can be learned efficiently in presence of Massart noise (before Diakonikolas’19 which is mentioned just before the above sentence) all achieved opt+epsilon type guarantees. These don't negate the point you want to get across since all 3 of these works are about a weaker model of corruption. You could add a qualification to you sentence that "Existing works on malicious noise ..." Coming to think of it, I think a more appropriate presentation of this work should cite the earlier Massart papers: Awasthi et al. COLT15 and COLT16.

Correctness: Seems correct.

Clarity: Ok

Relation to Prior Work: See above

Reproducibility: Yes

Additional Feedback:


Review 2

Summary and Contributions: This paper provides new theoretical results showing that the minimizer of the hinge loss (or other losses with similar properties) on a data set with a significant proportion of adversarially selected outliers can still have a small error on the inliers. These results are derived from a new analysis that holds whenever the inliers satisify a certain denseness criterion, which holds, for instance, for mixtures of isotropic log-concave distributions. The analysis can be applied to various settings, such as an adversarial flipping of a constant fraction of the labels, as well as adversatially selecting the location of each of the outliers. The results hold for a wider class of problems than previous works, which usually assume a unimodular distribution such as a single isotropic log-concave distribution. The analysis is clean and elegant, and relies on new definitions of data set properties which shed insight on the problem, and may be useful beyond the current work.

Strengths: The paper provides a novel and significant contribution to the study of learning with adversarial noise. The paper is well written, and insightful to read.

Weaknesses: I did not find any significant weaknesses. Minor suggestions are listed in the detailed comments section below.

Correctness: I verified the analysis in the body of the paper. Some of the proofs are in the supplementary material, which I did not fully verify.

Clarity: The paper is extremely well written.

Relation to Prior Work: Yes

Reproducibility: Yes

Additional Feedback: - In several locations, references are broken and question marks appear instead. - After equation (5), when reaching the contradiction, it is advisable to remind the reader the assumption on the RHS of equation (5) which leads to this conclusion. Similarly, after equation (9). - Remark 2: If f is non-convex, it does not have a sub-derivative. Thus, definition 4 requires some amendment to support such cases. - Remark 3: It is not clear to me how a validation set can be used to validate the pancake property, unless there is also indication of which data points in the validation set are inliers. ********* Authoer response: my minor concerns were sufficiently answered by the response. I keep my score as is.


Review 3

Summary and Contributions: The paper provides an analysis of the robustness to poisonous attacks of linear models learned with the large margin minimization problem. This analysis requires three set of assumptions: one set is on the true data distribution, which has to be log-concave and dense (even after applying linear transformations) and whose classes must be linearly separable with a margin; one set is on the contribution of the outliers to the loss, which has to be smaller than the relative contributions of inliers, and one on the nature of the considered loss function, which has to be L-admissible. ---- UPDATE ---- After reading the rebuttal and the other reviews, I increased my score to a plain accept.

Strengths: The main strength of the paper is in providing a theoretical analysis of large-margin linear classifiers allowing to lower bound the classification error by a value smaller than the corruption rate. Moreover, the assumption on the nature of the outliers is quite loose and can be applied to many scenarios considering outliers or label noise at training.

Weaknesses: The assumptions on the data distribution are limiting. In particular, the assumption that the data is margin separable does not stand in real case scenarios. The significance of the contribution could be improved by considering that a ratio of inliers lie within the margin and some degree of class overlapping. Another weakness is the lack of insights or intuitions on the results, especially for the bound of the HerSumNorm(O) of Theorem 13 which is hard to interpret. Finally, it is not clear to me how this analysis can be extended to non-linear classifiers, for instance using the kernel trick. Would the dense-pancake assumption stand still in the projected space? Could the dependence on the number of dimensions of the space be relaxed?

Correctness: The theoretical analysis seems correct. No empirical evaluation is provided.

Clarity: The paper is well written, although it lacks sometimes of rigor (see for instance Definition 6). It is not clear from the paper if the dense-pancake condition is novel. Also, a description of the adversarial label noise setting should be presented earlier in the introduction to avoid confusions with other adversarial settings mainly considered in recent years.

Relation to Prior Work: A direct comparison with other theoretical bounds cited as related works is missing. It would help in assessing the significance of the contributions to provide a comparison for instance on toy problems.

Reproducibility: Yes

Additional Feedback: - The paper has many typos and missing latex references. - From the title, one could think that the contributions are specific to the hinge loss, but they are for any admissible loss function and plugged in the large-margin risk minimization problem.


Review 4

Summary and Contributions: This paper shows that for learning a linear classifier, if the input distribution has constant margin, and satisfies the Dense Pancake assumption, then the hinge loss optimization tolerates a constant level of adversarial noise. Moreover, if the adversary is only allowed to perturb the label (not the input) of examples, then the margin condition can be relaxed to be as small as C/sqrt(d). This result shows that a mild margin condition guarantees a strong robustness property for the hinge loss optimization.

Strengths: This is a novel theoretical result. It makes two major contributions. First, it shows that the linear classifier learner can tolerate a much higher level of adversarial noise under the margin assumption and the Dense Pancake assumption. Second, it shows that on the original distribution, the classifier can achieve an error rate much lower than the noise level. Both results are interesting and can serve as good complementaries to the existing literature.

Weaknesses: According to the proof, the 4th assumption in Definition 4 is essential. It prohibits the theory to be directly applied to popular smooth loss functions like the logistic loss. Is there a way to relax this assumption?

Correctness: The theoretical result is correct.

Clarity: The paper is very well written. The concepts and terminologies are clearly defined, and the proof is elegantly sketched with good intuition. It is easy to follow the technical part of the paper.

Relation to Prior Work: yes

Reproducibility: Yes

Additional Feedback:

[Author Response · NeurIPS 2020]

We are grateful to the reviewers for their insightful reviews and feedback. We have incorporated fixes to simple issues such as typos and missing references and do not address those issues here.

Reviewer 1 pointed out that some of the works on Massart noise get to error $OPT + \epsilon$, under Massart Noise. We thank the reviewer for this correction and will update the introduction as the reviewer suggested, as well as add citations to the earlier works in the Massart noise model.

We thank the Reviewer 2 for pointing out the issue with defining subderivative without convexity. We will address this and change it to a suitable generalized subdifferential.

Reviewer 2 is correct in pointing out that using the validation set to validate the pancake property is subtle. We were indeed imagining access to a clean validation set, and will clarify this in the revision.

Reviewer 3 points out that the margin separable assumption may be limiting. We will discuss this more in the revised version: the margin is needed only for the inliers, and one can treat all points violating the margin assumption as outliers from the point of view of the analysis. In particular, this implies that the results hold as long as the fraction of (true outliers + inliers violating the margin assumption) is suitably small.

Reviewer 3 asked for intuition on the HerSumNorm in Theorem 13. We will add more intuition in the full version on how it shows up in the proof. Our goal for Theorem 13 was to use the weakest condition under which we could give a simple proof. For specific noise and data models, we can bound the HerSumNorm using standard techniques, as we do in the Supplementary material.

Reviwer 3's question on Kernel methods is a very interesting one. While our Theorem 13 would extend to Kernel spaces, our current approach to proving that for a random sample $D$, the pair $(D, \mu)$ satisfies the dense pancakes condition (in the Supplementary material) requires $\Theta(d)$ samples. While we can use random projections to $\Theta(1/\gamma^2)$ dimensions and apply the algorithm and the Theorem there, extending these results to the usual SVM with Kernels is a very compelling open question.

Reviewer 3 asks if the Dense Pancakes condition is novel. To our knowledge, this condition has not explicitly appeared in any previous work, though it has likely shown up implicitly as a step in similar results under strong distributional assumptions. We thank the reviewer for their other suggestions for improvement and will revise the paper accordingly.

Reviewer 4 asks if assumption 4 is necessary or can be relaxed to accommodate smooth losses such as the logistic loss. This is a very interesting suggestion, to which we can give two responses. The first is that one can define a (huberized version of the) loss $\max(f(y\mathbf{w}^\top \mathbf{x}), f(1))$ for $f$ being the logistic loss. Such a loss will satisfy the conditions and be close enough to the logistic loss for many purposes. The second response is that the proof can likely be extended to the actual logistic loss by adding an additional assumption on the HerSumNorm of the inliers. The place where we use assumption 4 is to derive the bound on the contribution from $I_2$ in (8). Suppose instead of condition (4) we had an upper bound, say $\frac{L}{(1+e)} < \frac{L}{3}$ on $|f'(x)|$ for $x \geq 1$. Then we could prove an upper bound of $\frac{L}{3} \cdot LinSumNorm(I_2) \leq \frac{L}{3} \cdot HerSumNorm(I)$ instead of 0 in equation (8). This would allow us to prove a variant of Thm 13 with an additional condition. This version will still imply a version of Thm 1 with slightly worse constants but now holding also for the logistic loss. We believe this additional complexity does not belong in the main theorem, but given the importance of the logistic loss, it would make sense to add such a statement in the Supplementary material, and we will do so in the revision. We thank the reviewer for this insightful question.

[Meta-Review · NeurIPS 2020]

All the reviewers find the results interesting and worth publishing. The minor issues they raised in their reviews were satisfactorily addressed by the rebuttal. I recommend accepting the paper.